# Revised Injury Severity Classification II (RISC II) is a predictor of mortality in REBOA-managed severe trauma patients

**Peter Hibert-Carius**[1], **David T. McGreevy**[2], **Fikri M. Abu-Zidan**[3]*, **Tal M. Hörer**[2], **the ABO-Trauma Registry Research Group**¶

**1** Department of Anesthesiology, Emergency and Intensive Care Medicine, Bergmannstrost Hospital Halle, Halle, Germany, **2** Department of Cardiothoracic and Vascular Surgery, Faculty of Medicine and Health, Örebro University, Örebro, Sweden, **3** Department of Surgery, College of Medicine and Health Science, UAE University, Al-Ain, United Arab Emirates

☯ These authors contributed equally to this work.
¶ Membership of the ABO-Trauma Registry Research Group is listed in the Acknowledgments.
* fabuzidan@uaeu.ac.ae

**Data Availability Statement:** All relevant data are within the manuscript and its Supporting Information files.

## Abstract

The evidence supporting the use of Resuscitative Endovascular Balloon Occlusion of the Aorta (REBOA) in severely injured patients is still debatable. Using the ABOTrauma Registry, we aimed to define factors affecting mortality in trauma REBOA patients. Data from the ABOTrauma Registry collected between 2014 and 2020 from 22 centers in 13 countries globally were analysed. Of 189 patients, 93 died (49%) and 96 survived (51%). The demographic, clinical, REBOA criteria, and laboratory variables of these two groups were compared using non-parametric methods. Significant factors were then entered into a backward logistic regression model. The univariate analysis showed numerous significant factors that predicted death including mechanism of injury, ongoing cardiopulmonary resuscitation, GCS, dilated pupils, systolic blood pressure, SPO2, ISS, serum lactate level and Revised Injury Severity Classification (RISCII). RISCII was the only significant factor in the backward logistic regression model (p < 0.0001). The odds of survival increased by 4% for each increase of 1% in the RISCII. The best RISCII that predicted 30-day survival in the REBOA treated patients was 53.7%, having a sensitivity of 82.3%, specificity of 64.5%, positive predictive value of 70.5%, negative predictive value of 77.9%, and usefulness index of 0.385. Although there are multiple significant factors shown in the univariate analysis, the only factor that predicted 30-day mortality in REBOA trauma patients in a logistic regression model was RISCII. Our results clearly demonstrate that single variables may not do well in predicting mortality in severe trauma patients and that a complex score such as the RISC II is needed. Although a complex score may be useful for benchmarking, its clinical utility can be hindered by its complexity.

**Funding:** The authors received no specific funding for this work.

**Competing interests:** The authors have declared that no competing interests exist.

**Abbreviations:** ABO, Aortic Balloon Occlusion; GCS, Glasgow Coma Scale; ISS, Injury Severity Score; REBOA, Resuscitative Endovascular Balloon Occlusion of the Aorta; RISCII, Revised Injury Severity Classification II.

## Introduction

Resuscitative Endovascular Balloon Occlusion of the Aorta (REBOA) is a minimally invasive procedure being increasingly used to prevent exsanguination in non-compressible torso haemorrhage in trauma patients and to bridge the time to definitive surgery for bleeding control. Control of bleeding using REBOA in both experimental and clinical studies was associated with improvement in coronary and cerebral perfusion with a possible positive survival benefit [1, 2]. In a single-center retrospective study, it was shown that early use of REBOA benefits patients with non-compressible abdominal or pelvic bleeding [3]. A recent study comparing 117 matched pairs of trauma patients from the Japanese National Trauma Registry found that survival significantly improved among severely injured trauma patients treated with REBOA compared with those treated without REBOA (45.3% compared with 32.5%) [4]. In contrast, another study using data from the American College of Surgeons Trauma Quality Improvement Program demonstrated a higher mortality rate in severely injured trauma patients who underwent REBOA placement compared with those who did not [5]. Other studies could not demonstrate survival benefits of using REBOA in trauma patients [6, 7]. We have recently shown that early use of hospital REBOA improves survival in patients presenting with pre-hospital traumatic cardiac arrest [8]. This indicates that the evidence supporting the use of REBOA in severely injured patients is still limited and needs more investigation. The ABO (Aortic Balloon Occlusion) Trauma Registry, which is collecting data on the use of REBOA in trauma patients having hemorrhagic shock, started on 2014. Data of the registry are retrieved from 22 centers of 13 countries from four continents (Europe, Asia, Africa, and South America). Using data from this registry, we aimed to define factors affecting mortality in trauma patients who underwent REBOA placement.

## Methods

### Ethical considerations

Ethical approval for the registry was obtained from the regional committee (study number: 2014/210; Regionala Etikprövningsnämnden, Uppsala, Sweden). Patient's data are anonymized at the point of registration with a unique registry-generated ID number. No patient identifiable data (name, hospital number, date of birth) are held in the registry and all data are held on a secure electronic database. Secured passwords have been given to centers joining the registry to be able to enter data. The registry is designed in line with the current European data protection regulation and conducted according to the principles expressed in the Declaration of Helsinki. Ethical approval of the current study was waived by the Ethical Committee of the Medical Association Saxony-Anhalt, Germany.

### Data retrieval

Data from the ABO (Aortic Balloon Occlusion) Trauma Registry, which were entered between 2014 and 2020, were analyzed. The ABOTrauma Registry contains data concerning trauma patients in whom REBOA was applied to treat haemorrhagic shock but does not include patients in whom REBOA was attempted but failed. Twenty-two centers from 13 countries participate in this registry. Almost 80% of the data were retrieved from four countries; Japan (33%), Columbia (23%), Russia (12%), and Italy (11%). This Registry contains both retrospective and prospective data. The centers contributing to data collection increased overtime. Centers which are known to use REBOA in clinical practice were invited to participate. Alternatively, they could register directly into the registry website and get their secure password after having approval from the investigators. To assure the generalizability of the data,

there were no limitations for hospital size or case volume. The registry is located and funded by the Department of Cardiothoracic and Vascular Surgery at Örebro University Hospital/ Sweden. The minimum data set of the registry includes 202 variables. Twenty-eight variables were used in this study; fifteen to calculate the RISC II (**S1 Appendix**); four were related to REBOA application (location, time, intermittent use, and partial occlusion), and the rest nine were deemed important factors related to resuscitation. Data were first retrieved and analysed on February 2020.

## Inclusion/exclusion criteria

Inclusion criteria for the present study were complete data to calculate the probability of survival using the updated Revised Injury Severity Classification (RISC II), availability of outcome data (30 days mortality), and REBOA performed within the first 2 hours after hospital admission.

## Calculations

The data needed to calculate the RISC II are summarised in **S1 Appendix**. The RISC II is currently the trauma score with the best prediction of mortality. The first version of the RISC score has been developed using data of 2 008 patients from the German Trauma Registry (TR-DGU) between 1993 and 2000 [9]. With improvements in trauma care, the observed mortality in the TR-DGU has fallen to about 2% below the prognosis of the RISC. An increasing number of cases did not receive a RISC score due to missing data [10]. An update of the Revised Injury Severity Classification score, the RISC II, has been developed using 30 866 patients from the TR-DGU between 2010 and 2011 which was validated with 21 918 patients from the same registry from 2012 [10]. An algorithm for replacing missing values in most of the used variables had been established [10].

## Statistical methods

The patients were divided into two groups: those who survived beyond 30 days and those who died within 30 days of admission (30 days mortality). Univariate nonparametric statistical methods were used for comparing these groups because these methods compare the ranks and not the crude numbers without the need for a normal distribution of the data.

The Mann-Whitney U test was used to compare ordinal or continuous data while Fisher's exact test was used to compare categorical data from these two groups. Significant factors having a *p* value of less than 0.05 in the univariate analysis were then entered into a backward logistic regression model to define factors affecting mortality.

A receiving operating characteristic (ROC) curve was used to define the area under the curve and the optimum cut-off value for the best predictors of mortality in the REBOA trauma patients as shown by the logistic regression model. Following the definition of the best cut point value having the best sensitivity and specificity, the number of true positive (TP), true negative (TN), false positive (FP) and false negative (FN) in predicting survival or death were calculated. The definitions which were used in predicting survival or death were as follows. A 'true positive' (TP) result was a positive outcome which was confirmed by a positive test. A 'false-positive' (FP) result was a negative outcome that was not confirmed by a positive test. A 'true-negative' (TN) result was a negative outcome that was confirmed by a negative test. A 'false-negative' (FN) result was a positive outcome that was not confirmed by a negative test. Accordingly, positive predictive value (PPV), negative predictive value (NPV), and usefulness index were calculated manually as follows: PPV = TP/(TP+FP), NPV = TN/(TN + FN), and usefulness index = sensitivity x (sensitivity—(1—specificity)). A test is regarded as useful if the

usefulness index is 0.35 or more [11, 12]. PASW Statistics 26, SPSS Inc., USA was used for analyzing the data. A p-value of less than 0.05 was accepted as significant.

## Results

During the study period, 253 patients that underwent REBOA were registered in the ABO-Trauma Registry. Of these, 189 (74.7%) had sufficient data to calculate the probability of survival by the RISC II and were included in this study. The majority of the patients were males (71.4%) and the median (range) age of all patients was 46 (4–96) years. The 30-day mortality was 49% (93 patients died) while 51% (96 patients) survived. The RISC II predicated a survival of 59% patients.

Table 1 shows the univariate analysis comparing demography and clinical data on arrival to the emergency department of REBOA trauma patients who survived (n = 96) and those who died (n = 93). Those who died had significantly more blunt trauma (p = 0.001, Fisher's Exact test), significantly more continuous CPR on arrival to the emergency room (p<0.0001, Fisher's Exact test), significantly lower blood pressure before inflation of the REBOA balloon

**Table 1. Univariate analysis comparing demography and clinical data of REBOA trauma patients who survived and those who died on arrival to emergency department.**

| Variable | Survived (n = 96) | Died (n = 93) | P* |
|---|---|---|---|
| Age (years) | 41 (15–88) | 50 (4–96) | 0.06 |
| Gender | | | 0.99 |
| Male | 69 (71.9%) | 66 (71%) | |
| Female | 27 (28.1%) | 27 (29%) | |
| Mechanism of injury | | | 0.001 |
| Blunt trauma | 67 (70.5%) | 83 (90.2%) | |
| Penetrating trauma | 28 (29.5%) | 9 (9.8%) | |
| Ongoing CPR | 2/90 (2.2%) | 17/85 (20%) | < 0.0001 |
| Heart rate (bpm) | | | 0.26 |
| 0 | 1 (1.1%) | 9 (11.5%) | |
| < 50 | 0 (0%) | 2 (2.6%) | |
| 50–100 | 24 (27.3%) | 18 (23.1%) | |
| 101–119 | 15 (26.1%) | 15 (19.2%) | |
| >120 | 40 (45.5%) | 34 (43.6%) | |
| SBP (mmHg) before inflation | 60 (0–150) | 50 (0–147) | 0.04 |
| Previous disease | 23/88 (26.1%) | 29/83 (34.9%) | 0.25 |
| Dilated pupils | 2/68 (2.9%) | 13/59 (22%) | 0.002 |
| ER intubation | 35/91 (38.5%) | 41/88 (46.6%) | 0.29 |
| Balloon location | | | 0.41 |
| Zone I | 71 (76.3%) | 74 (82.2%) | |
| Zone II | 4 (4.3%) | 1 (1.1%) | |
| Zone III | 18 (19.4%) | 15 (16.7%) | |
| Inflation time (min) | 36 (7–75) | 36 (6–75) | 0.76 |
| Partial inflation | 44/81 (54.3%) | 33/75 (44%) | 0.2 |
| Intermittent inflation | 19/64 (29.7%) | 21/57 (36.8%) | 0.44 |

Previous disease = American Society of Anesthesiologists (ASA) classification II or higher.

Data are presented as median (range) or number (%) as appropriate.

bpm: beats per minute, SBP: systolic blood pressure.

p = Fisher's Exact test or Mann-Whitney U test as appropriate.

(p = 0.04, Mann Whitney U test), and significantly more dilated pupils (p = 0.002, Fisher's Exact test).

**Table 2** compares severity and laboratory markers on arrival to the emergency department of the REBOA trauma patients who survived and those who died. Those who died had significantly higher injury severity scores (p<0.0001, Mann-Whitney U test), and serum lactate levels (p<0.0001, Mann-Whitney U test). They had significantly lower RISCII (p<0.0001, Mann-Whitney U test), GCS (p = 0.002, Mann-Whitney U test), oxygen saturation (p<0.0001, Mann-Whitney U test) and blood haemoglobin (p = 0.003, Mann-Whitney U test).

The logistic regression model was highly significant (p < 0.0001, Nagelkerke R2 = 0.35). **Table 3** shows the backward logistic regression model defining significant factors predicting mortality of REBOA trauma patients. RISCII was the only significant factor in this model (p < 0.0001). The odds of survival increased by 4% for each 1% increase in the RISCII.

**Fig 1** shows the receiver operator curve (ROC) of RISCII as a predictor of survival. The best RISCII that predicts 30-days survival in the REBOA treated patients is 53.7%, having a sensitivity of 82.3%, specificity of 64.5%, positive predictive value of 70.5%, negative predictive value of 77.9%, and usefulness index of 0.385. The ROC has an area under the curve of 80.2%. ISS was weaker as a predictor of death as it had an area under the curve of 74.3%.

## Discussion

Our study has shown that the most significant factor that predicted 30-day mortality in REBOA multiple trauma patients was RISCII. The odds of survival increased by 4% for each 1% increase in the RISCII. A RISCII of more than 54% predicted survival in these patients.

It is important to classify trauma patients according to injury severity and survival probability in order to make correct decisions based on clinical findings and point-of-care laboratory results that are available in the emergency room. Developing simple bedside scores that predict

**Table 2. Univariate analysis comparing severity and laboratory markers of REBOA trauma patients who survived and those who died on arrival to emergency department.**

| Variable | Survived (n = 96) | Died (n = 93) | P* |
|---|---|---|---|
| Injury Severity Score | 29 (11–75) | 48 (16–75) | <0.0001 |
| RISC II (predicated survival) | 84.6% (2.2–99.6%) | 37.8% (0.2–97.3%) | <0.0001 |
| Glasgow Coma Scale | | | 0.002 |
| 13–15 | 26 (57.8%) | 19 (28.8%) | |
| 9–12 | 8 (17.8%) | 14 (21.2%) | |
| 4–8 | 5 (11.1%) | 16 (24.2%) | |
| 3 | 6 (13.3%) | 17 (25.8%) | |
| SPO2 (%) | | | < 0.0001 |
| <80 | 8 (12.7%) | 20 (32.8%) | |
| 80–89 | 9 (14.3%) | 18 (29.5%) | |
| 90–100 | 46 (73%) | 23 (37.7%) | |
| Serum lactate (mmol/l) | 5.5 (1–16.2) | 9.6 (2–22.6) | < 0.0001 |
| Base deficit | -8.1 (-26 to 11.5) | -11 (-28.2 to 26) | 0.053 |
| Blood haemoglobin level (g/l) | 109 (29–181) | 93 (38–159) | 0.003 |
| Total Packed RBC | 10 (0–80) | 12 (0–143) | 0.37 |
| Total FFP | 8 (0–150) | 10 (0–70) | 0.98 |

Data are presented as median (range) or number (%) as appropriate.

*p* = Mann-Whitney U test.

**Table 3. Backward logistic regression model defining significant predictors for mortality of REBOA trauma patients (n = 189).**

|  | B | SE | Wald | p-value | OR | Lower limit 95% CI | Upper limit 95% CI |
|---|---|---|---|---|---|---|---|
| RISCII predicated survival (%) | -0.04 | .006 | 39.19 | <0.0001 | 0.96 | 0.95 | 0.97 |
| SBP before insertion (mmHg) | -0.007 | .006 | 1.48 | 0.22 | 0.99 | 0.98 | 1.00 |
| Constant | 2.59 | .493 | 27.64 | <0.0001 | 13.37 |  |  |

RISCII = Revised Injury Severity Classification, SBP = systolic blood pressure.

survival in early management of trauma patients is a very challenging task [13]. Although scores using multiple factors like RISCII is useful for comparing institutions, trauma systems and benchmarking, their utility in making decisions on individual patients in early

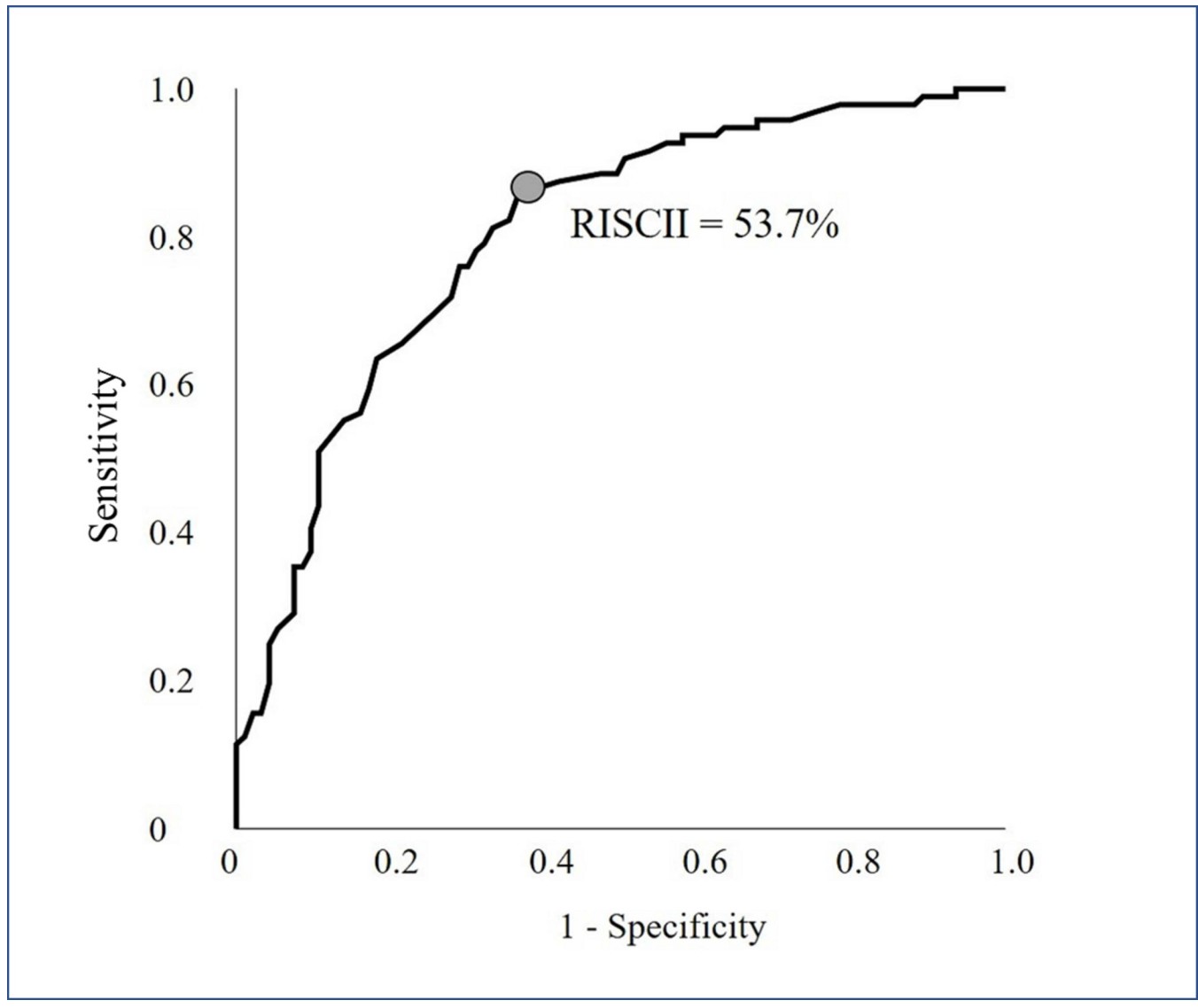

**Fig 1. The receiver operator curve (ROC) of RISCII as predictor of survival of REBOA trauma patients (n = 189) treated at 22 centers in 13 countries during the period 2014–2020.** The best point of RISCII for predicting 30-day mortality is 53.7%.

management of severely injured patients are limited and hindered by their complexity. This is expected to be even more difficult in patients in extremis such as REBOA patients who may have a wide range of variation in their response to life-threatening bleeding.

By analyzing the ABOTrauma Registry data, we could identify various factors affecting mortality in the univariate analysis. Patients with higher ISS, low GCS, dilated pupils, and blunt trauma had a higher mortality, which is similar to other reported studies [10, 14–16]. Shock is a generalized status of body tissue hypoperfusion that leads to lactic acidaemia. Hypotension, high serum lactate, and low blood haemoglobin level was significantly more in those who died compared with those who survived in our study similar to others [14–16]. REBOA inflation time, location, partial inflation or intermittent inflation did not have a significant effect on mortality in the current study. Although this is true in our study, we have to stress that our sample size is small, the inflation time was relatively short (a median of 35 minutes) and almost identical in those who died and those who survived. However, while bleeding control was achieved, REBOA may have caused ischemia reperfusion injury leading to delayed multiple organ failure [17, 18]. The predictive value of vital signs, such as heart rate, has been questioned in numerous studies [10, 19–21]. Our findings confirm that heart rate, which is not included in the survival calculation model of the RISC II, is of little value in predicting mortality in trauma patients.

REBOA was inflated in Zone II in 5 patients (2.6%) which is contraindicated. REBOA patients of the ABOTrauma registry are all emergency patients in a time critical situation with a high number of blind catheter insertion. Therefore, an unintentionally placement in zone II especially with the older devices without proper length marking is not surprising.

The only significant factor that remained in the logistic regression was the RISC II. The results of this study highlight the ability of RISC II to predict survival in the severely exsanguinating patients who have a very narrow window of survival. The results also demonstrate that single variables may not successfully predict mortality in severe trauma patients due to the complexity of the situation. A complex score such as the RISC II is therefore needed. It is reassuring that the RISC II, which was developed and tested for patients not receiving REBOA, did well in predicting mortality in a selected REBOA group, which validates this score. Although this can be useful for comparing trauma systems and benchmarking, its clinical utility is limited and hindered by its complexity.

RISC II replaced ISS in the logistic regression model when both were in the same model because it was stronger and because of the colinearity effect, since both of them are derived from the same anatomical injury severity scores. Accordingly, the best predictor will stay in the model if both are included.

## Limitations

Our study has certain limitations that should be highlighted. First, one major concern is the small sample size. This is shown in some of the variables that have a trend for statistical difference (like the age and base deficit) which is possibly caused by type II statistical error. Increased age is a strong predictor of mortality [9, 10] but this could not be demonstrated in our study. Second, our registry includes patients who had successful REBOA insertion and does not include patients in whom REBOA was attempted and failed. Since we cannot report the data of this important group, a high risk of selection bias exists. Third, we had no institutional control on the patients included, which poses a risk of selection bias. Fourth, the partial retrospective nature of the first period of our registry has affected the completeness of the data, 25% of the registry were excluded due to missing data. We decided not to impute these missing data by replacing them with the average of each variable otherwise our confidence in the

analysis would then be much weaker. Systematic error and bias are very relevant in this scenario. This is a common problem when studying patients with high risk of mortality. Fifth, we have used the 30-day in hospital mortality as our main outcome variable. This is justified on the assumption that mortality in REBOA patients is mainly related to bleeding. This acceptable approach was used by other major trauma registries [22, 23]. Nevertheless, we have to acknowledge that about 5% of the deaths in trauma patients occur after 30 days [24] and it would have been better to use the in-hospital mortality. Sixth, blunt trauma was more associated with death in our study, however, a more detailed description of the injury pattern is unfortunately not available, as AIS was not captured in the data apart from the head injury. It is important to avoid this major limitation in the future development of the ABOTrauma Registry. Seventh, it is very important to stress that our data, which is from 22 centers in 13 different countries and 4 continents collected over 6 years, is heterogeneous. We can for example observe the very wide range of age of the study population because the registry included both children and adults. Heterogeneity is unavoidable in such situation, particularly when reporting real-world practice. This is not bad per se as it reflects the generalizability of the data. Finally, the Adjusted R Squared values of our final model was low (0.35). This indicates that this model explains less than 40% of data variation. Other important explanatory variables were, therefore, not captured. Missing variables is a common limitation when patients have a very high mortality.

## Conclusions

Predicting mortality in severely exsanguinating multiple trauma patients who underwent REBOA placement is a challenging task. Although there are multiple significant factors shown in the univariate analysis, the only factor that predicted 30-day mortality in REBOA multiple trauma patients in a logistic regression was RISCII. The best point to predict survival in these patients was a RISCII of more than 54%. Our results clearly demonstrate that single variables may not do well in predicting mortality in severe trauma patients and that a complex score such as the RISCII is needed. Although a complex score may be useful for benchmarking, its clinical utility can be hindered by its complexity.

## Supporting information

**S1 Appendix. Variables included in the model of the RISC II [10].**
(DOCX)

**S1 File. Data of the study as Excel file.**
(XLSX)

## Acknowledgments

Professor Tal M. Hörer is the leader of the ABO-Trauma Registry research group (tal.horer@-regionorebrolan.se).

**Contributors (ABOTrauma Registry research group)**
M. Sadeghi[2], A. Pirouzram[2], A. Toivola[2], P. Skoog[4], Y. Matsumura [5], M. Maszkowski[6], M. Falkenberg[7], S. W. Chang[8], B. Kessel[9], G. Shaked[10], M. Bala[11] F. Coccolini[12], K. F. Nilsson[2], V. Reva[13]

**Affiliations**
[2] Department of Cardiothoracic and Vascular Surgery, Faculty of Medicine and Health, Örebro University, Örebro, Sweden.

[4] Department of Hybrid and Interventional Surgery, Unit of Vascular Surgery, Sahlgrenska University Hospital, Gothenburg, Sweden.

[5] Department of Emergency and Critical Care Medicine, Chiba University Graduate School of Medicine, Chiba, Japan.

[6] Västmanlands Hospital Västerås, Department of Vascular Surgery, Örebro University, Örebro, Sweden.

[7] Department of Radiology, Sahlgrenska University Hospital, Gothenburg, Sweden.

[8] Department of Thoracic and Cardiovascular Surgery, Trauma Center, Dankook University Hospital, Cheonan, Republic of Korea.

[9] Department of Surgery, Hillel Yaffe Medical Centre, Hadera, Israel.

[10] Department of Anesthesiology and Critical Care, Soroka University Medical Center, Ben Gurion University, Beer Sheva, Israel.

[11] Trauma and Acute Care Surgery Unit, Hadassah Hebrew University Medical Center, Jerusalem, Israel.

[12] Department of Surgery, Papa Giovanni XXIII Hospital, Bergamo, Italy.

[13] Department of War Surgery, Kirov Military Medical Academy, Saint Petersburg, Russia.

## Author Contributions

**Conceptualization:** Peter Hibert-Carius, David T. McGreevy, Fikri M. Abu-Zidan, Tal M. Hörer.

**Data curation:** David T. McGreevy, Tal M. Hörer.

**Formal analysis:** Peter Hibert-Carius, Fikri M. Abu-Zidan.

**Methodology:** Peter Hibert-Carius, David T. McGreevy, Fikri M. Abu-Zidan.

**Supervision:** Fikri M. Abu-Zidan, Tal M. Hörer.

**Validation:** Peter Hibert-Carius.

**Writing – original draft:** Peter Hibert-Carius, David T. McGreevy, Fikri M. Abu-Zidan.

**Writing – review & editing:** Peter Hibert-Carius, David T. McGreevy, Fikri M. Abu-Zidan, Tal M. Hörer.

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
