## [Decision Letter · Decision Letter 0]

2 Nov 2020

PONE-D-20-30062

Factors affecting mortality in REBOA-managed severe trauma patients

PLOS ONE

Dear Dr. Abu-Zidan,

Thank you for submitting your manuscript to PLOS ONE. After careful consideration, we feel that it has merit but does not fully meet PLOS ONE’s publication criteria as it currently stands. Therefore, we invite you to submit a revised version of the manuscript that addresses the points raised during the review process.

We look forward to receiving your revised manuscript.

Kind regards,

Zsolt J. Balogh, MD, PhD, FRACS

Academic Editor

PLOS ONE

Journal Requirements:

2. Please provide more details on data collection, including the date(s) on which you accessed the databases or records to obtain the data used in your study, what information you collected, and data was necessary to determine Inclusion/ Exclusion.

4. One of the noted authors is a group or consortium [ABO-Trauma Registry research group]. In addition to naming the author group and listing the individual authors and affiliations within this group in the acknowledgments section of your manuscript, please also indicate clearly a lead author for this group along with a contact email address.

Reviewers' comments:

Reviewer's Responses to Questions

**Comments to the Author**

1. Is the manuscript technically sound, and do the data support the conclusions?

Reviewer #1: Yes

Reviewer #2: Yes

2. Has the statistical analysis been performed appropriately and rigorously? 

Reviewer #1: I Don't Know

Reviewer #2: I Don't Know

3. Have the authors made all data underlying the findings in their manuscript fully available?

Reviewer #1: Yes

Reviewer #2: Yes

4. Is the manuscript presented in an intelligible fashion and written in standard English?

Reviewer #1: Yes

Reviewer #2: Yes

5. Review Comments to the Author

Reviewer #1: GENERAL COMMENTS

The use of registry-based real-World data to further inform on the role of REBOA in the injured patient is very welcome. It appears from this study that the previously validated RISC II scoring system is reasonably accurate in predicting outcomes in REBOA patients once a predicted score of 54% is reached. Further clarification is needed regarding the utility of this finding and the confidence in using it to informing decision-making for injured patients.

SPECIFIC COMMENTS

INTRODUCTION

P3L59 ‘surgical bleeding control’ may not be a term well-known to some readers – suggest replacing with time to ‘definitive surgery for bleeding (or haemorrhage) control’ or similar

P3L60 – ‘improvement in coronary and cerebral perfusion pressure’ – were these physiological observations demonstrated in human studies or animal studies? This statement is embedded in a sentence combined with the statement of ‘possible positive survival benefit’ implying human studies. Suggest being more clear surrounding the evidence supporting REBOA in terms of separating pre-clinical animal studies and human hospital / registry data findings.

P3L63 – ‘A recent study……survival significantly improved’. Again more background on what kind of study this was is required.

P3L68 – ‘could not ALSO demonstrate’ – suggest removing ALSO

P3L72 – ‘ABOTrauma Registry’ – a line or two here about what this registry is, describe acronym here for the first time rather than in the Methods section, to complement further details given in Methods section

METHODS

Nowhere in the Methods section is there a clear description of which variables are being studied. We are told that ‘Data from the ABOTrauma registry were analyzed’ and that ‘Inclusion criteria were complete data to calculate probability of survival using….RICS II’.

Therefore the reader is left wondering were ALL the data points from the ABOTrauma registry analysed? If so how many are there and what exactly are they? Or was it just the variables that make up the RISCII score? However if this is the case when we come to look at the results there are variables reported that are not part of RISC II, such as SPO2, total packed RBC, total FFP, but also an absence of reported variables that do comprise RISC II, such as worse and second worst injury by AIS, or motor function (total GCS instead is presented). Therefore it is unclear what the authors initially set out to study - this needs complete clarity with an unambiguous description of the variable studied from the outset and the rationale explaining them.

P3L97 - What is the definition of ‘REBOA performed early after admission’. ‘Early’ is a vague and ambiguous description of time without definition.

P4L90 – More detail about the ABOTrauma registry is required, where is it hosted / administered / based – an abridged version of that provided within reference [8] would be ideal. A large number of centres contribute from numerous countries all with presumably varying levels of experience and expertise with REBOA. In order for the reader to get a richer idea of the types of healthcare systems / % blunt vs penetrating trauma in each population etc contributing to the registry, could the authors please consider providing the principal 3-5 countries (or all of them) that contribute to the registry and what is each country’s % contribution to the database? I would guess that the top 3-5 countries would account for the majority, or at least over 50%, of the database entries?

P4L90 – Furthermore additional clarity is required regarding the registry - this is important because the reference given [8] regarding the ABOTrauma registry refers to a registry comprising of 11 centres from 7 countries between 2014-2019, whereas this manuscript, which states is using the same registry within a very similar timeframe (2014-2020), is using a registry of 22 centres from 13 countries. Thus the same registry appears be comprised of very different contributors across both manuscripts. If the additional 1 year of data capture in 2020 accounts for this difference then the data in 2020 will presumably significantly skew the data, unless the additional 11 centres and 6 countries all have very similar REBOA practices?

P5L104 – Although discussed in the Discussion, a brief line or two is required here regarding RISC II and why it was chosen – what is the proven utility thus far in the published literature? In a previous paper the authors state that in their opinion it is the best trauma score for predicting outcomes currently available. Please furnish this manuscript with a similar level of detail and justification.

P5L108 – Why was 30-day mortality chosen rather than in-hospital mortality?

RESULTS

P5L123 – ‘253 patients’ suggest adding ‘253 that underwent REBOA’ or ‘253 REBOA patients’ or similar

P5L123 - What is the denominator? Need to know the what % of the total registry the REBOA patients comprised

P5L124 – 25.3% of REBOA patients were excluded due to lack of data. This raises two points worthy of further comment in the Discussion please: (i) the RISC II model requires some data points that are not readily immediately available by the bedside (laboratory results, pre-injury health status can be vague etc) and some have argued that this represents a weakness of such scoring systems that require many data points – the larger and more predictive they are, the more unwieldy and less user-friendly they become. A scoring system that is unusable in a quarter of a population from a well-established and well-maintained international trauma registry is likely to be far less useful in the real-World setting outside of trauma registries. (ii) The second point is more relevant to this paper and is touched upon in the discussion but needs further explanation and that is regarding the management of missing data. The RISC II authors spent a lot of time exploring how best to manage missing data and in fact one of the key messages from the updated RISC II model was that missing values are not excluded or assigned, but in fact included. Therefore why have the authors decided to exclude data from a model designed and validated to handle such inconsistencies?

P6L126 – The age range of the patients is vast (4-96). If we assume that the age range presented in Table 1 is in fact IQR and not total range (it is potentially incorrectly labelled as range) – this is still 15 – 88. Such wide age ranges suggest a very heterogeneous group and I wonder how comparable they are. I am surprised that the authors have not made some discussion regarding age trending towards being a predictor of mortality (P=0.06) in the initial comparison.

P6L131 – Blunt trauma is significantly associated with mortality compared to penetrating. Presumably this is a reflection of multi-organ polytrauma that normally accompanies blunt mechanisms? However there is no presentation of ISS / AIS or similar for the author to make this association. Please provide / discuss.

P7 Table 1 – 5.4% of REBOA inflations are in Zone 2 – an area conventionally contraindicated for inflation within the visceral segment of the aorta. No discussion is provided on this an is required. Were these inadvertent inflations?

P7 Table 1 – Explain what the variable ‘previous disease’ encompasses. As it stands it is vague and difficult to know how discrimatory it can be as a variable.

P7L139 – “Data presented as the median (range)”. The age ranges differ from text (4-96) to table (15-88). Presumably IQR are being presented in the table but it is not stated as such.

P9 Table 3 – Why present ‘SBP before insertion’ if this is not significant

P9L170 – Remove ‘This is the’ and ‘Title’ from Fig 1

P9L172 - Explain how the best point for predicting 30-day mortality is 53.7% is usuable clinically.

P10L174 – RISC II was removed from the analysis. There is no explanation in the Methods that this was planned or the statistical or methodological reason for doing so. Please justify

P10 Table 4 – Why present ‘SBP before insertion’ and ‘Blood haemoglobin level’ if these were not significant

DISCUSSION

P10L184 – Would add to this statement that the most important factor ‘that was measured’….

P10L189 – Do the authors use ISS and survival probability models to make decisions for individual patients as implied here? The next sentence implies that such predictive models are used for clinical decision making at the most challenging time of assessment of a trauma patient’s care when they are in extremis and one is working with limited information. Clear explanation needs to given here regarding the utility of such scoring systems and models – the implication is that they can be used at an individual level on admission to the ER – rather than their more established role in comparing institutions, trauma systems and benchmarking.

P10L190 – Do you mean ‘in extremis’ rather than ‘in extremes’

P11L195 – The statement regarding shock status needs to be clearer - It is stated that base deficit was higher in those that died – whilst there is a non-significant trend it is discussed in combination with factors that were found to be significant such as SBP and lactate.

P11L196 – The statement regarding REBOA inflation time not having a significant impact on mortality needs to be made in combination with highlighting that the times were almost identical in both groups, in addition to the comment that inflation time was relatively short. I have commented on Zone 2 inflation above and I suspect numbers are too small to make any meaningful comparison. Similarly partial / intermittent inflation times are roughly similar in both groups. Whilst stating that none of these had any significant impact on mortality is correct within this study, such statements need to be put in the context of the data from which they are gleaned.

P11L202- Are pain, anxiety and beta-blocker use really thought to be the most important factors of variations in heart rate in hypotensive trauma patients, rather than haemodynamic drivers due to hypovolaemia?

P11L207 – Does a specificity of 82% and sensitivity of 64% once a RISC II score of 53% is reached warrant the phrase ‘extreme strength’ for survival predictability? How do these figures compare to other established trauma predictive models / scoring systems such as RISC II for all-comers (not just REBOA), TRISS, ISS, wISS etc). How does a ROC AUC of 80.2% compare with other models?

LIMITATIONS

Suggest removing italic numbers throughout

P12L222 – When discussing sample size type 2 errors should be discussed for some of the results that demonstrated trends but failed to reach statistical significance

P12L225 – Need to discuss the variation in trauma epidemiology / trauma networks and systems / type (% blunt) / pre-hospital care / REBOA practices etc across the 22 centres and 13 different countries. This will complement the suggestions made above regarding explaining the registry in more detail in the Methods section. Heterogeneity is not necessarily a bad thing, particularly when informing real-World practice, and unavoidable in such settings, but this needs discussing.

P12L227 – What is meant by the group in whom ‘intention to insert REBOA’ failed? Is this a group who arrested pre-hospital, or a technical failure to gain access and deliver the device? This is a potentially interesting subgroup.

P12L230 – See earlier comments regarding the discordance between how missing data was managed in the present study and the inclusion of missing data in the original description of the RISC II model (Ref 10). Please explain why the authors have high confidence in the validated RISC II model where missing data is included, but confidence in the current analysis with inclusion of missing data would be weak.

P12L230 – remove the semicolon (;)

DISCUSSION

P13L243 – Clarify and simplify the clinical significance of the findings. Is it that if a severely injured trauma patient has a RISC II score of 54% or more then RISC II is useful to predict survival if REBOA is used? Does this mean that if their RISC II score is 54% the utility of the scoring system in this scenario is only slightly better than tossing a coin - but gets more predictive the higher the number (the worse the predicted survival is)? If possible please put this into clinical context – what does a RISC II score of 54% look like clinically – perhaps correlate with ISS to aid readers understand how this fit’s with their trauma cohort. Perhaps in the REBOA population the majority of patients have a very high predicted mortality anyway and therefore loss of prediction in the lower % cohorts is clinically negligible?

SUPPORTING INFORMATION

P20L352 – Remove ‘This is the S1 Table Title’

Reviewer #2: Nicely written work.

Please find some comments/questions below.

1. you state data collection methods are described in Ref 8, but I was unable to identify a summary description outlining what data is actually collected in the ABO Trauma Registry. making this information clearly available might be relevant given the rather noticeable number of patients excluded and therefore the relevance of this aspect of your methods.

2. in your inclusion criteria you say "early" after admission. if possible defining this somehow more precisely would benefit the manuscript.

3. your mentioned exclusion criterion is just the opposite of your first inclusion criteria and seems redundant.

4. typographic error page 7 line 147, GCC.

5. The selection bias that this work carries, yet acknowledged as a second limitation by the authors, possibly remains the main concern of this work. Any additional information that might tame this, might lessen the readership's concerns on the manuscript.

6. why the need for the second logistic regression model presented in table 4?

7. adding standard paragraphs to the abstract may add clarity to the manuscript.

8. the manuscript reads in a way that it seems that since inception ( i.e. the chosen selection criteria) this work was aimed at validating the RISCII score rather then defining aspects influencing care in REBOA candidates, it that is indeed the case the authors might want to consider rewording their title and the focus of their introduction accordingly.

6. PLOS authors have the option to publish the peer review history of their article (what does this mean?). If published, this will include your full peer review and any attached files.

Reviewer #1: No

Reviewer #2: No

---

## [Author Response · Author response to Decision Letter 0]

7 Dec 2020

7th December 2020

Professor Zsolt J. Balogh, 

Academic Editor

PLOS ONE

Ref: PONE-D-20-30062: Factors affecting mortality in REBOA-managed severe trauma patients

Dear Professor Balogh

Thank you for your prompt response and giving us the chance to re-submit a revised version of the above manuscript. The manuscript has now been completely re-written as advised by the reviewers. We thank the reviewers for their highly encouraging and constructive comments which have significantly improved our manuscript. All changes made in the manuscript are highlighted in yellow color to facilitate the review process. The answers to the reviewers’ comment are as follows:

Reviewer #1: 

GENERAL COMMENTS

The use of registry-based real-World data to further inform on the role of REBOA in the injured patient is very welcome. It appears from this study that the previously validated RISC II scoring system is reasonably accurate in predicting outcomes in REBOA patients once a predicted score of 54% is reached. Further clarification is needed regarding the utility of this finding and the confidence in using it to informing decision-making for injured patients.

Answer: Thank you for your useful general advice. These two important points will be covered in detail later in the specific comments.

SPECIFIC COMMENTS

INTRODUCTION (OK)

Comment 1: P3L59 ‘surgical bleeding control’ may not be a term well-known to some readers – suggest replacing with time to ‘definitive surgery for bleeding (or haemorrhage) control’ or similar

Answer: The sentence has been modified as requested (Page 3, Line 60).

Comment 2: P3L60 – ‘improvement in coronary and cerebral perfusion pressure’ – were these physiological observations demonstrated in human studies or animal studies? This statement is embedded in a sentence combined with the statement of ‘possible positive survival benefit’ implying human studies. Suggest being more clear surrounding the evidence supporting REBOA in terms of separating pre-clinical animal studies and human hospital / registry data findings.

Answer: Thank you for your comment. This sentence has been more clarified as requested indicating that evidence exists in both experimental and clinical studies. (P3, L61).

Comment 3: P3L63 – ‘A recent study……survival significantly improved’. Again more background on what kind of study this was is required.

Answer: Thank you for your comment. More details on the study have been added as requested (P3, L65).

Comment 4: P3L68 – ‘could not ALSO demonstrate’ – suggest removing ALSO

Answer: “also” was deleted as requested.

Comment 5: P3L72 – ‘ABOTrauma Registry’ – a line or two here about what this registry is, describe acronym here for the first time rather than in the Methods section, to complement further details given in Methods section

Answer: Description of the registry has been added as requested (P3, L74). 

METHODS

Comment 6: Nowhere in the Methods section is there a clear description of which variables are being studied. We are told that ‘Data from the ABOTrauma registry were analyzed’ and that ‘Inclusion criteria were complete data to calculate probability of survival using….RICS II’.

Therefore the reader is left wondering were ALL the data points from the ABOTrauma registry analysed? If so how many are there and what exactly are they? Or was it just the variables that make up the RISCII score? However if this is the case when we come to look at the results there are variables reported that are not part of RISC II, such as SPO2, total packed RBC, total FFP, but also an absence of reported variables that do comprise RISC II, such as worse and second worst injury by AIS, or motor function (total GCS instead is presented). Therefore it is unclear what the authors initially set out to study - this needs complete clarity with an unambiguous description of the variable studied from the outset and the rationale explaining them.

Answer: Thank you for your important comment. You are completely right. The RICS II calculation methodology permits to calculate the value despite having some missing data. The registry has 202 variables in which we used only 28 variables. The variables that were selected are those that are needed to calculate the RICS II and those that can be available in the early stages of the management of the patients with severe bleeding. This has now been clarified (P5, L106). 

Comment 7: P3L97 - What is the definition of ‘REBOA performed early after admission’. ‘Early’ is a vague and ambiguous description of time without definition.

Answer: Thank you for this comment which we agree with. Early REBOA is REBOA which was performed within the first 2 hours after hospital admission. This has now been clarified as requested (P5, L114).

Comment 8: P4L90 – More detail about the ABOTrauma registry is required, where is it hosted / administered / based – an abridged version of that provided within reference [8] would be ideal. A large number of centres contribute from numerous countries all with presumably varying levels of experience and expertise with REBOA. In order for the reader to get a richer idea of the types of healthcare systems / % blunt vs penetrating trauma in each population etc contributing to the registry, could the authors please consider providing the principal 3-5 countries (or all of them) that contribute to the registry and what is each country’s % contribution to the database? I would guess that the top 3-5 countries would account for the majority, or at least over 50%, of the database entries?

Answer: More details about the registry have been added as requested (P4, L98). Almost 80% of the data were supplied by four countries; Japan (33 %), Columbia (23 %), Russia (12 %), and Italy (11 %). This has now been added to the methods section. 

Comment 9: P4L90 – Furthermore additional clarity is required regarding the registry - this is important because the reference given [8] regarding the ABOTrauma registry refers to a registry comprising of 11 centres from 7 countries between 2014-2019, whereas this manuscript, which states is using the same registry within a very similar timeframe (2014-2020), is using a registry of 22 centres from 13 countries. Thus the same registry appears be comprised of very different contributors across both manuscripts. If the additional 1 year of data capture in 2020 accounts for this difference then the data in 2020 will presumably significantly skew the data, unless the additional 11 centres and 6 countries all have very similar REBOA practices?

Answer: Thank you for your kind comment and correct observation. The centers are almost the same. We have to clarify that the countries mentioned in reference 8 are those which contributed to the data of that specific study while in this study data were retrieved from all centers. 

Comment 10: P5L104 – Although discussed in the Discussion, a brief line or two is required here regarding RISC II and why it was chosen – what is the proven utility thus far in the published literature? In a previous paper the authors state that in their opinion it is the best trauma score for predicting outcomes currently available. Please furnish this manuscript with a similar level of detail and justification.

Answer: Thank you regarding the comment and the interest in the RISC II. More details on RISCII have been added as requested (P5, L117).

Comment 11: P5L108 – Why was 30-day mortality chosen rather than in-hospital mortality?

Answer: Thank you for your kind comment. We have clarified why we used the 30 day mortality and discussed this issue in the limitations of the study (P13, L270). 

RESULTS

Comment 12: P5L123 – ‘253 patients’ suggest adding ‘253 that underwent REBOA’ or ‘253 REBOA patients’ or similar

Answer: done (P7, L155) 

Comment 13: P5L123 - What is the denominator? Need to know the what % of the total registry. the REBOA patients comprised

Answer: The registry is only for REBOA patients. This has now been clarified in the limitations of the study (P13, L261).

Comment 14: P5L124 – 25.3% of REBOA patients were excluded due to lack of data. This raises two points worthy of further comment in the Discussion please: (i) the RISC II model requires some data points that are not readily immediately available by the bedside (laboratory results, pre-injury health status can be vague etc) and some have argued that this represents a weakness of such scoring systems that require many data points – the larger and more predictive they are, the more unwieldy and less user-friendly they become. A scoring system that is unusable in a quarter of a population from a well-established and well-maintained international trauma registry is likely to be far less useful in the real-World setting outside of trauma registries. (ii) The second point is more relevant to this paper and is touched upon in the discussion but needs further explanation and that is regarding the management of missing data. The RISC II authors spent a lot of time exploring how best to manage missing data and in fact one of the key messages from the updated RISC II model was that missing values are not excluded or assigned, but in fact included. Therefore why have the authors decided to exclude data from a model designed and validated to handle such inconsistencies?

Answer: Thank you for your important precise comment and your raised concern. Point I has been highlighted in the discussion (P11, L218). Point II, The corresponding author (FAZ), who is a senior statisticain believes that data should not be imputed and missing data not to be replaced in the analysis because it cannot be assumed that missing data was random especially in those who died. Nevertheless, the first author is from the German school and his well validated approach to calculate missing data is respected and incorporated. This data has been tested in large cohort of patients compared with the small size of the current study. It is important to accomadte other opinion if it was acceptable as occured in this case. Furthermore, the RISC II was developed with the data set of the TR-DGU and some of the parameters used to replace missing data are not captured in the ABOTrauma registry and therefore we weren’t able to calculate the RISC II for all patients in the registry, when values of the core parameters for the RISC II were missing. 

Comment 15: P6L126 – The age range of the patients is vast (4-96). If we assume that the age range presented in Table 1 is in fact IQR and not total range (it is potentially incorrectly labelled as range) – this is still 15 – 88. Such wide age ranges suggest a very heterogeneous group and I wonder how comparable they are. I am surprised that the authors have not made some discussion regarding age trending towards being a predictor of mortality (P=0.06) in the initial comparison.

Answer: Thank you for your concern. We have checked the data. The median (range) for the study population (all patients) is correct which is 46 (4-96) years. Those in the table are for the two seperate groups (those who died and those who survived). We have now clarified this in the text (P7 , L158). The range is wide because the registry included both children and adults. This wide range reflects the heterogeneity of data which can be the cause of the non-significant difference between the groups in age. Nevertheless, these results with their heterogeneity represent “real life” data. This has now been added to the limitation section of the study (P14, L278). 

Comment 16: P6L131 – Blunt trauma is significantly associated with mortality compared to penetrating. Presumably this is a reflection of multi-organ polytrauma that normally accompanies blunt mechanisms? However there is no presentation of ISS / AIS or similar for the author to make this association. Please provide / discuss.

Answer: Blunt trauma was more associated with death, however, a more detailed description of the injury pattern is unfortunately not available, as AIS was not captured in the data apart from head injury. This is an important part of its future development ABOTrauma Registry. This has now been added to the limitations of the study (P13, L275).

Comment 17: P7 Table 1 – 5.4% of REBOA inflations are in Zone 2 – an area conventionally contraindicated for inflation within the visceral segment of the aorta. No discussion is provided on this an is required. Were these inadvertent inflations?

Answer: We agree that the balloon was wrongly inflated in this region. REBOA patients of the ABOTrauma registry are all emergency patients in a time critical situation with a high number of blind catheter insertion. Therefore an unintentionally placement in zone II especially with the older devices without proper length marking is not surprising. This has now been discussed as advised (P12, L240).

Comment 18: P7 Table 1 – Explain what the variable ‘previous disease’ encompasses. As it stands it is vague and difficult to know how discrimatory it can be as a variable.

Answer: Thank you for your comment. This has been defined in the footnote of the table as American Society of Anesthesiologists (ASA) classification II or higher (P8, L172).

Comment 19: P7L139 – “Data presented as the median (range)”. The age ranges differ from text (4-96) to table (15-88). Presumably IQR are being presented in the table but it is not stated as such.

Answer: Similar to point 15. 

Comment 20: P9 Table 3 – Why present ‘SBP before insertion’ if this is not significant

Answer: Thank you for your kind query. We have used backward logistic regression in the analysis which will keep the variables that best explain the data with the highest R squared. Even if the SBP is not significant, keeping it in the model will give the best model for these data. The corresponding author is a senior statistician who is the Statistical Editor of both World J Emerge Surgery (IF 4.1) and European J Emerg Surg (IF 1.9). We can assure the reviewer that the model presented in table 3 is proper and should be kept as it is.

Comment 21: P9L170 – Remove ‘This is the’ and ‘Title’ from Fig 1

Answer: done (P11, L206) 

Comment 22: P9L172 - Explain how the best point for predicting 30-day mortality is 53.7% is useable clinically. 

Answer: We have now discussed the utility of this finding in more detail as requested (P11, L218). Furthermore, we have also calculated more relevant clinical variables including the positive predictive value and the negative predictive value which depend on the prior probability of death in the studied population. In addition, we calculated the usefulness index and referenced it. This index test is useful if it is more than 0.35 which was the case here. (Methods, P6, L141; and results P10 , L203)

Comment 23: P10L174 – RISC II was removed from the analysis. There is no explanation in the Methods that this was planned or the statistical or methodological reason for doing so. Please justify

Answer: Thank you for highlighting this important point. We agree with reviewer 2 that this was not needed and this step was removed.

Comment 24: P10 Table 4 – Why present ‘SBP before insertion’ and ‘Blood haemoglobin level’ if these were not significant

Answer: Table 4 has been deleted (see comment 23)

DISCUSSION

Comment 25: P10L184 – Would add to this statement that the most important factor ‘that was measured’….

Answer: The sentence has now been corrected as advised (P11, L211). 

Comment 26: P10L189 – Do the authors use ISS and survival probability models to make decisions for individual patients as implied here? The next sentence implies that such predictive models are used for clinical decision making at the most challenging time of assessment of a trauma patient’s care when they are in extremis and one is working with limited information. Clear explanation needs to given here regarding the utility of such scoring systems and models – the implication is that they can be used at an individual level on admission to the ER – rather than their more established role in comparing institutions, trauma systems and benchmarking.

Answer: We agree with this very important and accurate comment. This has now been rephrased and discussed to highlight the above point (P11, L218).

Comment 27: P10L190 – Do you mean ‘in extremis’ rather than ‘in extremes’

Answer: corrected (P11, L223).

Comment 28: P11L195 – The statement regarding shock status needs to be clearer - It is stated that base deficit was higher in those that died – whilst there is a non-significant trend it is discussed in combination with factors that were found to be significant such as SBP and lactate.

Answer: Thank you for your observation. The sentence has now been rephrased to be more accurate reflecting our own results (P11, L228).

Comment 29: P11L196 – The statement regarding REBOA inflation time not having a significant impact on mortality needs to be made in combination with highlighting that the times were almost identical in both groups, in addition to the comment that inflation time was relatively short. I have commented on Zone 2 inflation above and I suspect numbers are too small to make any meaningful comparison. Similarly partial / intermittent inflation times are roughly similar in both groups. Whilst stating that none of these had any significant impact on mortality is correct within this study, such statements need to be put in the context of the data from which they are gleaned.

Answer: Thank you for your comment. This section has been rephrased as advised (P12, L231).

Comment 30: P11L202- Are pain, anxiety and beta-blocker use really thought to be the most important factors of variations in heart rate in hypotensive trauma patients, rather than haemodynamic drivers due to hypovolaemia?

Answer: Thank you for your comment. This statement has been deleted.

Comment 31: P11L207 – Does a specificity of 82% and sensitivity of 64% once a RISC II score of 53% is reached warrant the phrase ‘extreme strength’ for survival predictability? How do these figures compare to other established trauma predictive models / scoring systems such as RISC II for all-comers (not just REBOA), TRISS, ISS, wISS etc). How does a ROC AUC of 80.2% compare with other models?

Answer: Thank you for this highly important point. We have now compared the performance of RISC II compared with ISS. ISS had an area under the curve of 74.3% (P10, L205).

The wording has been rephrased to reduce the tone and to be realistic. It reads now as “The results of this study highlight the ability of RISC II to predict survival in the severely exsanguinating patients who have a very narrow window of survival” (P12, L244). 

LIMITATIONS

Comment 32: Suggest removing italic numbers throughout

Answer: done

Comment 33: P12L222 – When discussing sample size type 2 errors should be discussed for some of the results that demonstrated trends but failed to reach statistical significance

Answer: Thank you for this important point. Type II statistical error for some of the results that showed strong trend has been discussed in the limitations of the study as advised (P13, L258).

Comment 34: P12L225 – Need to discuss the variation in trauma epidemiology / trauma networks and systems / type (% blunt) / pre-hospital care / REBOA practices etc across the 22 centres and 13 different countries. This will complement the suggestions made above regarding explaining the registry in more detail in the Methods section. Heterogeneity is not necessarily a bad thing, particularly when informing real-World practice, and unavoidable in such settings, but this needs discussing.

Answer: Thank you for this very important point which has now been highlighted in the limitations section (P14, L278). 

Comment 35: P12L227 – What is meant by the group in whom ‘intention to insert REBOA’ failed? Is this a group who arrested pre-hospital, or a technical failure to gain access and deliver the device? This is a potentially interesting subgroup.

Answer: Thank you for this important comment. We do agree that there are some cases in which REBOA wasn´t successful. Unfortunately data on them were not collected. To address this important concern we have now clarified in the methods section that the ABOTrauma registry includes only patients with successfully placed REBOAs and does not include data of failed REBOA attempts (P4, L96) and discussed this issue in the limitations of the study (P13 , L261).

Comment 36: P12L230 – See earlier comments regarding the discordance between how missing data was managed in the present study and the inclusion of missing data in the original description of the RISC II model (Ref 10). Please explain why the authors have high confidence in the validated RISC II model where missing data is included, but confidence in the current analysis with inclusion of missing data would be weak.

Answer: Similar to point 14.

Comment 37: P12L230 – remove the semicolon (;)

Answer: done 

DISCUSSION

Comment 38: P13L243 – Clarify and simplify the clinical significance of the findings. Is it that if a severely injured trauma patient has a RISC II score of 54% or more then RISC II is useful to predict survival if REBOA is used? Does this mean that if their RISC II score is 54% the utility of the scoring system in this scenario is only slightly better than tossing a coin - but gets more predictive the higher the number (the worse the predicted survival is)? If possible please put this into clinical context – what does a RISC II score of 54% look like clinically – perhaps correlate with ISS to aid readers understand how this fit’s with their trauma cohort. Perhaps in the REBOA population the majority of patients have a very high predicted mortality anyway and therefore loss of prediction in the lower % cohorts is clinically negligible?

Answer: Similar to comments 22 and 26, the answer is detailed there.

SUPPORTING INFORMATION

Comment 39: P20L352 – Remove ‘This is the S1 Table Title’

Answer: done

Reviewer #2: 

Comment 1. Nicely written work, you state data collection methods are described in Ref 8, but I was unable to identify a summary description outlining what data is actually collected in the ABO Trauma Registry. making this information clearly available might be relevant given the rather noticeable number of patients excluded and therefore the relevance of this aspect of your methods.

Answer: Thank you for your highly encouraging comment (Similar to point 6 of reviewer 1). 

Comment 2. in your inclusion criteria you say "early" after admission. if possible defining this somehow more precisely would benefit the manuscript.

Answer: Similar to comment 7 of reviewer 1.

Comment 3. your mentioned exclusion criterion is just the opposite of your first inclusion criteria and seems redundant.

Answer: Thank you for your comment. Redundant sentence was deleted.

Comment 4. typographic error page 7 line 147, GCC.

Answer: GCC corrected to GCS (P9, L181).

Comment 5. The selection bias that this work carries, yet acknowledged as a second limitation by the authors, possibly remains the main concern of this work. Any additional information that might tame this, might lessen the readership's concerns on the manuscript.

Answer: This has been rearranged and highlighted as requested to be the major concern of the study (P13, L258).

Comment 6. why the need for the second logistic regression model presented in table 4?

Answer: Thank you for your comment. We agree that this was not needed and the table was deleted.

Comment 7. adding standard paragraphs to the abstract may add clarity to the manuscript.

Answer: Thank you for highlighting this important issue. We completely agree that a structured abstract is very useful, but we had to follow the format of the journal.

Comment 8: the manuscript reads in a way that it seems that since inception ( i.e. the chosen selection criteria) this work was aimed at validating the RISCII score rather then defining aspects influencing care in REBOA candidates, it that is indeed the case the authors might want to consider rewording their title and the focus of their introduction accordingly.

Answer: Thank you for your suggestion to modify the title to reflect our findings. We have now modified the title but we did not change the introduction because we aimed to study other predictors besides the RISCII score in the study population.

Editorial comments

1. The manuscript meets PLOS ONE's style requirements,

2. Professor Tal M Hörer is the leader of the ABO-Trauma Registry research group (tal.horer@regionorebrolan.se). This has now been added to the first page and acknowledgment section.

3. Availability of data and material: Data used in this study will be made public after acceptance of this paper.

4. Ethics statements is now only included in the methods section as requested.

5. The anonymous used data are attached as excel sheet.

6. More details on the time data were retrieved and analyzed are added to the manuscript as requested (Page 5, first paragraph)

Thank you for your consideration for this revised manuscript. We hope that these changes will satisfy the reviewers and that the manuscript finally finds a place in your reputable journal.

Yours Sincerely

Fikri Abu-Zidan

---

## [Decision Letter · Decision Letter 1]

30 Dec 2020

PONE-D-20-30062R1

Revised Injury Severity Classification II (RISC II) is a strong predictor of mortality in REBOA-managed severe trauma patients

PLOS ONE

Dear Dr. Abu-Zidan,

Thank you for submitting your manuscript to PLOS ONE. After careful consideration, we feel that it has merit but does not fully meet PLOS ONE’s publication criteria as it currently stands. Therefore, we invite you to submit a revised version of the manuscript that addresses the points raised during the review process.

We look forward to receiving your revised manuscript.

Kind regards,

Zsolt J. Balogh, MD, PhD, FRACS

Academic Editor

PLOS ONE

Additional Editor Comments (if provided):

Dear Authors,

The comment "strong" in the title is a bit to much. I recommend dropping it and refer to RIC II as "..is a predictor of mortality in REBOA-managed ...".

Reviewers' comments:

Reviewer's Responses to Questions

**Comments to the Author**

1. If the authors have adequately addressed your comments raised in a previous round of review and you feel that this manuscript is now acceptable for publication, you may indicate that here to bypass the “Comments to the Author” section, enter your conflict of interest statement in the “Confidential to Editor” section, and submit your "Accept" recommendation.

Reviewer #1: All comments have been addressed

Reviewer #2: All comments have been addressed

2. Is the manuscript technically sound, and do the data support the conclusions?

Reviewer #1: Yes

Reviewer #2: Yes

3. Has the statistical analysis been performed appropriately and rigorously? 

Reviewer #1: I Don't Know

Reviewer #2: Yes

4. Have the authors made all data underlying the findings in their manuscript fully available?

Reviewer #1: Yes

Reviewer #2: Yes

5. Is the manuscript presented in an intelligible fashion and written in standard English?

Reviewer #1: Yes

Reviewer #2: Yes

6. Review Comments to the Author

Reviewer #1: (No Response)

Reviewer #2: Thanks and well done for thoroughly and timely addressing all comments and concerns.

While the revised manuscript tones down the findings to adequately reflect results, it would be beneficial to avoid misinterpretation for a less experienced readership to highlight further the limited clinical value of the RISC II score for REBOA patients. In fact, while this concept is adequately addressed by the authors in the discussion (P11, L 218-224) , it could also be reiterated in page 12 L 248, "is therefore needed" ....for? and ideally also at the very end of the conclusion in the main mansucript and in the abstract "[...] needed."

7. PLOS authors have the option to publish the peer review history of their article (what does this mean?). If published, this will include your full peer review and any attached files.

Reviewer #1: **Yes: **Mr Joseph Dawson

Reviewer #2: No

---

## [Author Response · Author response to Decision Letter 1]

30 Dec 2020

Dear Professor Balogh

Thank you for your prompt response and giving us the chance to re-submit a revised version of the above manuscript. The manuscript has now been revised as advised. We thank the reviewers for their input which have significantly improved our manuscript. All changes made in the manuscript are highlighted in yellow color to facilitate the review process. The answers to the comments are as follows:

Editor 

Comment: "strong" in the title is a bit to much. I recommend dropping it and refer to RIC II as "..is a predictor of mortality in REBOA-managed ...".

Answer: Thank you for your kind advice. The title has been changed as advised.

Reviewer #2: 

Comment. Thanks and well done for thoroughly and timely addressing all comments and concerns. While the revised manuscript tones down the findings to adequately reflect results, it would be beneficial to avoid misinterpretation for a less experienced readership to highlight further the limited clinical value of the RISC II score for REBOA patients. In fact, while this concept is adequately addressed by the authors in the discussion (P11, L 218-224) , it could also be reiterated in page 12 L 248, "is therefore needed" ....for? and ideally also at the very end of the conclusion in the main mansucript and in the abstract "[...] needed."

Answer: Thank you for your useful recommendation. The statements have been rephrased as advised (Page 2, line 53; Page, 12 line 251; and Page 14, line 297). 

Thank you for your consideration for this revised manuscript. We hope that these changes will satisfy the reviewers and that the manuscript finally finds a place in your reputable journal.

Yours Sincerely

Fikri Abu-Zidan

---

## [Decision Letter · Decision Letter 2]

14 Jan 2021

Revised Injury Severity Classification II (RISC II) is a predictor of mortality in REBOA-managed severe trauma patients

PONE-D-20-30062R2

Dear Dr. Abu-Zidan,

We’re pleased to inform you that your manuscript has been judged scientifically suitable for publication and will be formally accepted for publication once it meets all outstanding technical requirements.

Kind regards,

Zsolt J. Balogh, MD, PhD, FRACS

Academic Editor

PLOS ONE

Additional Editor Comments (optional):

Congratulations Professor Abu-Zidan! Best Regards, Zsolt J. Balogh

Reviewers' comments:

Reviewer's Responses to Questions

**Comments to the Author**

1. If the authors have adequately addressed your comments raised in a previous round of review and you feel that this manuscript is now acceptable for publication, you may indicate that here to bypass the “Comments to the Author” section, enter your conflict of interest statement in the “Confidential to Editor” section, and submit your "Accept" recommendation.

Reviewer #1: All comments have been addressed

Reviewer #2: All comments have been addressed

2. Is the manuscript technically sound, and do the data support the conclusions?

Reviewer #1: Yes

Reviewer #2: Yes

3. Has the statistical analysis been performed appropriately and rigorously? 

Reviewer #1: Yes

Reviewer #2: I Don't Know

4. Have the authors made all data underlying the findings in their manuscript fully available?

Reviewer #1: Yes

Reviewer #2: Yes

5. Is the manuscript presented in an intelligible fashion and written in standard English?

Reviewer #1: Yes

Reviewer #2: Yes

6. Review Comments to the Author

Reviewer #1: (No Response)

Reviewer #2: (No Response)

7. PLOS authors have the option to publish the peer review history of their article (what does this mean?). If published, this will include your full peer review and any attached files.

Reviewer #1: No

Reviewer #2: No

---

## [Editor Report · Acceptance letter]

27 Jan 2021

PONE-D-20-30062R2 

Revised Injury Severity Classification II (RISC II) is a predictor of mortality in REBOA-managed severe trauma patients 

Dear Dr. Abu-Zidan:

I'm pleased to inform you that your manuscript has been deemed suitable for publication in PLOS ONE. Congratulations! Your manuscript is now with our production department. 

Kind regards, 

on behalf of

Dr. Zsolt J. Balogh 

Academic Editor

PLOS ONE